# Detailed analysis and comparison of different activity metrics

**Bálint Maczák**[1,ID], **Gergely Vadai**[1,ID,*], **András Dér**[2], **István Szendi**[3,4], **Zoltán Gingl**[1]

1 Department of Technical Informatics, University of Szeged, Szeged, Hungary, 2 Institute of Biophysics, Biological Research Centre, Eötvös Loránd Research Network, Szeged, Hungary, 3 Department of Psychiatry, Albert Szent-Györgyi Medical School, University of Szeged, Szeged, Hungary, 4 Psychiatry Unit, Kiskunhalas Semmelweis Hospital University Teaching Hospital, Kiskunhalas, Hungary

☯ These authors contributed equally to this work.
* vadaig@inf.u-szeged.hu

**Data Availability Statement:** Our analysis presented in this study is based on our measurement data, which is publicly available. The 42 10-days-long raw triaxial acceleration signals (measured on different healthy human subjects'

## Abstract

Actigraphic measurements are an important part of research in different disciplines, yet the procedure of determining activity values is unexpectedly not standardized in the literature. Although the measured raw acceleration signal can be diversely processed, and then the activity values can be calculated by different activity calculation methods, the documentations of them are generally incomplete or vary by manufacturer. These numerous activity metrics may require different types of preprocessing of the acceleration signal. For example, digital filtering of the acceleration signals can have various parameters; moreover, both the filter and the activity metrics can also be applied per axis or on the magnitudes of the acceleration vector. Level crossing-based activity metrics also depend on threshold level values, yet the determination of their exact values is unclear as well. Due to the serious inconsistency of determining activity values, we created a detailed and comprehensive comparison of the different available activity calculation procedures because, up to the present, it was lacking in the literature. We assessed the different methods by analysing the triaxial acceleration signals measured during a 10-day movement of 42 subjects. We calculated 148 different activity signals for each subject's movement using the combinations of various types of preprocessing and 7 different activity metrics applied on both axial and magnitude data. We determined the strength of the linear relationship between the metrics by correlation analysis, while we also examined the effects of the preprocessing steps. Moreover, we established that the standard deviation of the data series can be used as an appropriate, adaptive and generalized threshold level for the level intersection-based metrics. On the basis of these results, our work also serves as a general guide on how to proceed if one wants to determine activity from the raw acceleration data. All of the analysed raw acceleration signals are also publicly available.

non-dominant wrists, sampled at 10 Hz in the ±8 g measurement interval) are downloadable from Figshare under CC-BY 4.0 license through the following DOI: 10.6084/m9.figshare.16437684. The data is shared in binary format; therefore, we have attached a MATLAB function to read them with ease, its documentation can be found in the source file. The presented correlation analysis is performed on diversely calculated activity signals based on our measurement data. The resulting 148×148 correlation matrices are also publicly available in the S1 Table (XLSX) of the supporting information.

**Funding:** This research was supported by the Hungarian Government and the European Regional Development Fund under the grant number GINOP-2.3.2-15-2016-00037 ("Internet of Living Things"). The publication was supported by the University of Szeged Open Access Fund under the grant number 5461. The funders had no role in study design, data collection and analysis, decision to publish, or preparation of the manuscript.

**Competing interests:** The authors have declared that no competing interests exist.

## Introduction

Actigraphy is a widespread method utilized mainly in medicine, biophysics and sports science [1–6], but nowadays, it also tends to be popular in casual everyday use. The method applies a small, acceleration sensor-based, non-invasive diagnostic tool which is called actigraph, independently of the manufacturer. The actigraph, worn on the non-dominant wrist or sometimes on the hip, can objectively, reliably and cost-effectively record the locomotor activity of the subject. Based on the analysis of the measured acceleration signal, the actigraph generates an activity value for successive non-overlapping time slots, i.e. epochs of equal length, using an activity metric and stores the created activity signal in its memory. It is important to note that this activity signal is often referred to as physical activity in general, which is rather ambiguous because physical activity can also be an indicator obtained by further processing of the activity signal [7–9].

One of the most common fields of use of actigraphic recordings is sleep medicine since, by analysing the activity signal, scientists can reliably characterize the subjects' sleep quality [10] and possible disturbance in their circadian rhythm [11], which are commonly related to mental disorders [12]. Therefore, actigraphy is also frequently utilized in psychiatric research; for example, to recognize behavioural disorders or to distinguish between similar mental diseases [13]. Another immensely current use of actigraphs is to determine human physical activity (PA) [4]. In addition to their therapeutic usage, actigraphic devices are also successfully utilized in the field of human motion pattern analysis [5,6]; for example, to find regularities in the distribution of humans' resting and active period durations.

Since actigraphy is a widely and regularly applied multidisciplinary method, its high degree of standardization and generalization is required and expected. Conversely, multiple studies have already pointed out that the utilization of actigraphs is inconsistent, independently of the field of use. As a consequence, direct comparisons of findings between different studies and the exact reproduction of the results are problematic in several cases. In the following, we explain this in an extensive and general way.

The main reason of this issue is the lack of a standardized activity metric; in fact, multiple activity calculation procedures exist [14]. The activity metric describes the way the activity values are calculated from the somehow preprocessed acceleration signal. However, the activity calculation procedures are lossy data reduction methods that are implemented by closely guarded proprietary algorithms and vary by manufacturer. Therefore, the activity data obtained from different actigraph brands and models are derived from raw acceleration data in a disparate and generally incomparable way [14]. For example, the preprocessing step frequently includes signal filtering techniques, but the filter properties also depend on the manufacturer or the given device [7].

At first glance, one might also think that if we know the exact type of device, it is enough information about the activity determination procedure, but unfortunately, in multiple cases, it is false. Consider e.g., the manufacturer of one of the most commonly used actigraph devices in the field of physical activity research: ActiGraph. In their devices' documentations, there is no information about the parameters of their bandpass filtering method. To resolve the issue, an independent study tried to identify these parameters with reverse engineering [15]. The same is true for the exact way how these devices calculate their proprietary Activity Count (AC) metric. Similarly to ActiGraph, studies often name the activity values as "activity counts" in general, even if they are determined in diverse ways and therefore are not comparable to each other [3]. For another example, the most commonly used devices in sleep medicine have two intersection-based activity metrics, yet the exact determination of these threshold values is unclarified by the manufacturer.

Another issue is that not only the devices are lacking documentation, but actigraphy measurements in studies are not sufficiently reported either. Since scientists use these devices as turnkey solutions in an end-user manner, they often do not feel the necessity to report such details that would be crucial for the reader to understand how to interpret their unit of activity. For example, in a physical activity-related review article, the authors highlight that 80% of the studies they have examined do not report any information on the used filter [16]. Multiple studies further criticize the reporting complexity of studies, as scientists frequently incompletely describe the main aspects of the utilization of actigraphic devices, such as device type, data processing, activity metric, sampling rate, epoch length, software [17,18].

Not to mention that more and more commercially available actigraphic devices are able to store the raw triaxial acceleration data instead of activity values. If one decides to use such a device and still wants to analyse activity values, the whole activity determination process must be done by oneself after the measurement. Although these manufacturers attempt to give a helping hand in the form of software packages or documentation [19,20], they are often not compatible with each other, nor with those devices that immediately produce activity output. These conflicts further enhance the diversification of activity calculation procedures among scientists, which is contradictory to standardization.

The aim of this study is not only to highlight these issues again, but to examine the effect of the processing steps leading from the raw acceleration signals to the activity signals with the purpose to help scientists to explore, differentiate and choose between the diverse activity calculation possibilities, which is especially helpful if their device records the raw acceleration signal. In the following, we are summarizing those questions that generally arise during the activity value determination process.

The raw acceleration signal contains the gravity of Earth. If it is required to be eliminated, we can simply accomplish it by subtracting 1 g from the magnitudes of the acceleration vector then taking the absolute value of the difference [21], but this technique is not feasible in the case of axial data. However, a similar effect can be achieved by digital filtering. The most commonly used filtering technique involves a bandpass filter which additionally removes higher frequency components besides the low-frequency components, including the gravity of Earth. The filter can be applied on per axis or on the magnitudes of the acceleration [22–24]. Following from this, multiple datasets are generatable from a single triaxial acceleration signal depending on the steps performed in the preprocessing phase.

Since multiple activity metrics are widespread in the literature, we have to decide which metric we wish to use in the next step. Nevertheless, not any activity metric is applicable on all datasets. To make it even more complicated, in the case of numerous activity metric and dataset combinations, we can achieve correctly interpretable activity signals, even if their combination seems incompatible at first glance, by applying further corrections on the resulting activity signal. Commonly, several metrics can be applied on a specific dataset, and a metric can be used on multiple datasets. As a consequence, surprisingly many activity signals can be generated from a single measurement. This creates the necessity to examine the differences and relationships between both the obtained activity signals and the possible preprocessings. Moreover, some metrics can have unclarified technical aspects. For instance, the threshold values are not satisfactorily described in the literature in the case of the level intersection-based activity metrics.

Nowadays, it is almost self-evident to measure acceleration on three axes, but how do we handle the triaxial dataset when we determine the activity value? The activity metric can be applied to the vector magnitudes, but also on each axis individually [25]. If we do the latter, we somehow need to determine a single activity value from the three activity values obtained from the three axes for a given epoch.

As presented, there are various issues that are already marked by different studies [3,7,14,16–18]. Nonetheless, a detailed and comprehensive comparison of the activity metrics and possible preprocessings has been missing from the literature so far. Therefore, our goal was to map the relationship between the different activity determination methods by such a comprehensive analysis. For this purpose, we recorded the raw acceleration signals in three directions and then compared the activity signals calculated in different ways. Due to fast technological progress, it is easier and easier to develop or even access a device that measures and collects the subjects' raw triaxial acceleration with relatively high frequency for a long period of time, which opened up the way to digital postprocessing. Therefore, the collection and analysis of the impact of the activity calculation steps presented below also serve as a guide on what activity metrics to use, how to use them and how they relate to each other.

## Actigraphic recordings

We performed a large-sampled measurement on healthy human subjects equipped with actigraph devices specially developed for this project and placed on their non-dominant wrists. We intended to collect the human motion data in the most general way to ensure flexibility in the activity determination process and in further analyses based on the calculated activity signals.

We have developed a small device housed in a 3D-printed capsule of size 41 mm × 16 mm × 11.3 mm (LWH) which weighs only 5.94 grams. The device fits in a commercially available wristband, integrates a microcontroller (C8051F410), a 3-axis MEMS acceleration sensor (LIS3DH) and a 1GB-capacity flash memory chip. The high-performance acceleration sensor is intended to be used in motion sensing applications and provides both the user-programmable sampling rate and the accurate analogue-to-digital conversion. Since the microcontroller receives digital data from the sensor, the overall accuracy of the device is the same as the accuracy of the sensor. The detailed specifications are given in the freely available datasheet. The device incorporates a real-time quartz clock as well to ensure long-term timing accuracy with a tolerance of +/- 20 ppm.

The microcontroller continuously reads the data from the accelerometer chip, whose sampling rate can be set from 1 sample/sec to 100 samples/sec. At the used rate of 10 samples/sec, the device is capable of recording raw acceleration data continuously into the device's flash memory for more than three weeks. According to the above, our solution provides all the features required for the acquisition of the acceleration signals, just as the commercially available actigraphs. In addition, we have full control over the measurement process, all the specifications are openly accessible, and any kind of signal processing can be implemented.

With these devices, we carried out 84 measurements on different individuals from April to June, 2019. All the participants were recruited from the whole student community at the University of Szeged, Hungary. They were evaluated with the structured interviews of Structured Clinical Interview for DSM-5, Clinical Version (SCID-5-CV) [26] and Personality Disorder (SCID-5-PD) [27] for excluding any mental disorders. Volunteer students with a personal history of psychiatric disorder, history of head injury with loss of consciousness for more than 30 min, current substance abuse, or any medical illness that could significantly constrain neurocognitive functions were excluded. With this selection strategy, such a heterogenous group of healthy participants was generated that is easily reproducible, and reduces the possibility that extreme cases (for example, patients with mental disorders or epilepsy) impact our results.

The individual measurements were approximately 14 days long. The collected free-living acceleration signals were sampled at 10 Hz in the ±8 g measurement interval, as this configuration is viable in diverse fields of use [1,28–32], and our study is not restricted to any specific

discipline. For the analyses presented in this article, we selected 42 data series with the length of exactly 10 days. The aim of the selection was to pick those measurements where the subject rarely took off the device, and only for short periods (for example, when bathing). Since in this study, we only compare the resulting activity signals calculated from the measured acceleration data in a technical sense, and we are not performing any analysis based on activity signals, it was not necessary to filter out these short sections. The selected 42 subjects (all Caucasian, 23 female) were 18–25 years of age (mean 22.46, SD 1.86). The data series of their movements' raw triaxial accelerations are publicly available (10.6084/m9.figshare.16437684). We also published a MATLAB function to read these binary files with ease, which is attached to the shared data.

## Common procedures of determining activity values

In the diverse research area of actigraphy, many different methods are used to calculate activity values, but details of the applied methods are rarely shown. This makes it difficult to evaluate and compare results as well as to collect the methods of activity calculation [17,18].

The general steps of determining activity values are depicted in Fig 1. The 3-axis acceleration sampled at relatively high rate (1–100 Hz [1,33]) is preprocessed, which means filtering the signals, calculating the magnitude of acceleration and, if no filtering has taken place, normalizing. An activity value is then determined for each epoch of the signal based on an activity metric. The length of the epoch depends on the activity metrics and the area of use (e.g. physical activity measurement, sleep medicine, etc.), but it is generally between 1 s and 60 s. Consequently, the sampling time of the obtained activity signal is the length of the epoch.

Some metrics are applied to the axial vector components of the measured acceleration (uniaxial mode) thus generating activity signals per axis or one activity signal from a somehow defined composition of these. While other metrics are applied to the magnitude of acceleration vector calculated from the 3-axis vector components (triaxial mode), in this case only one activity signal is produced. Activity metrics are responsible for data reduction in actigraphs; their use is justified by the limited resources of the devices since in this way, it is not necessary to save the acceleration signal sampled with high rate in the memory of the device, only to store one activity value per epoch.

It can be seen in Fig 1 that in the case of the classic, widespread actigraphs, the whole sequence of operations is accomplished within the hardware. In the analysis presented below, our device was only responsible for the measurement and data storage sampled at 10 Hz; everything else could be done digitally after the measurement. Thus, the different activity

**Task of the hardware in classical solutions**

```
Measuring raw 3-axis          Preprocessing:        Different    Cutting  one  of  the      Determining    activity
acceleration data             • Filtering           datasets     dataset's data to T_e       values      for     each
                              • Normalization                    long epochs                 epoch  based  on  an
T_s = 0.01 s … 1 s            • Calculating magnitude                                        activity metric
                                of acceleration                  T_e = 1 s … 1 min
```

**Task of the hardware in the presented solution**

Uniaxial → ← Triaxial

Activity signal sampling time: $T_e$

**Fig 1. The general steps of determining activity values.** The activity for a given $T_e$ epoch length calculated from the 3-axis acceleration data sampled with $T_s$ sampling time.

determination methods and metrics became directly comparable. Before we introduce our examination methods, we are summarizing the different preprocessing steps and activity metrics which could be found in the literature so one can see what issues arise during preprocessing and activity calculation if we have digitized acceleration signals.

## Preprocessing

Depending on how the signals are preprocessed–e.g. whether filtering is applied or gravity of Earth (*g*) is removed in a different way (normalization), and whether applied to axial acceleration or the magnitude of acceleration–the same metric may give a different value. So, beyond metrics, it is equally important what input dataset we apply them to. Unfortunately, this is mentioned even less often in the literature. The possible dataset types that result from the preprocessing step are reviewed below with a nomenclature introduced by us.

Let UFXYZ be a dataset of vector components along the x, y, and z axes, referred to as UFX, UFY, UFZ per axis. These are the raw axial acceleration data read from the actigraph without any modification. An example of such signals can be seen in Figs 2A and 3.

Most actigraphic devices of the market include a bandpass filter to filter the signals provided by the acceleration sensor. The lower cut-off frequency of the bandpass filter is between 0.2 Hz and 0.5 Hz, and the upper cut-off frequency is between 2.5 Hz and 7.0 Hz. However, not all the necessary information can be found on the exact parameters of the used filters; for example, the order of the filter is often not indicated [15,16,34]. The use of a bandpass filter is explained by the fact that by filtering the low-frequency components, the DC component (i.e., in this case the *g*) can be eliminated. Using a high-pass filter, tremors and high-frequency noise can be filtered out [2]. Since the order of the digital filter used in one of the common devices has been identified as 3rd order [15,35], and based on this, other studies have already used a 3rd order Butterworth bandpass filter in the preprocessing of the acceleration signal, we have also used the same digital filter with 0.25 Hz and 2.5 Hz cut-off frequencies. To examine the effect of the filter characteristics, we also performed the tests with a 30th order filter with the same parameterization. Since the results obtained with the two filters showed the same correlation pattern for the activities calculated in different ways, the case of the 3rd order filter is

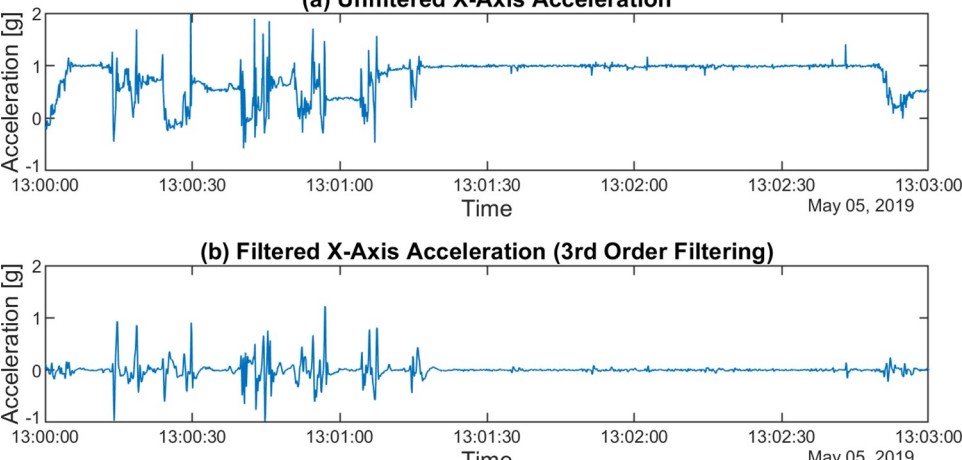

**Fig 2. The effect of different filtering methods.** The x-axis acceleration (a) was filtered using a 3rd order (b) digital Butterworth bandpass filter with fL = 0.25 Hz, fH = 2.5 Hz corner frequencies. The waveforms cover the same 3-minute-long section of a subject's motion.

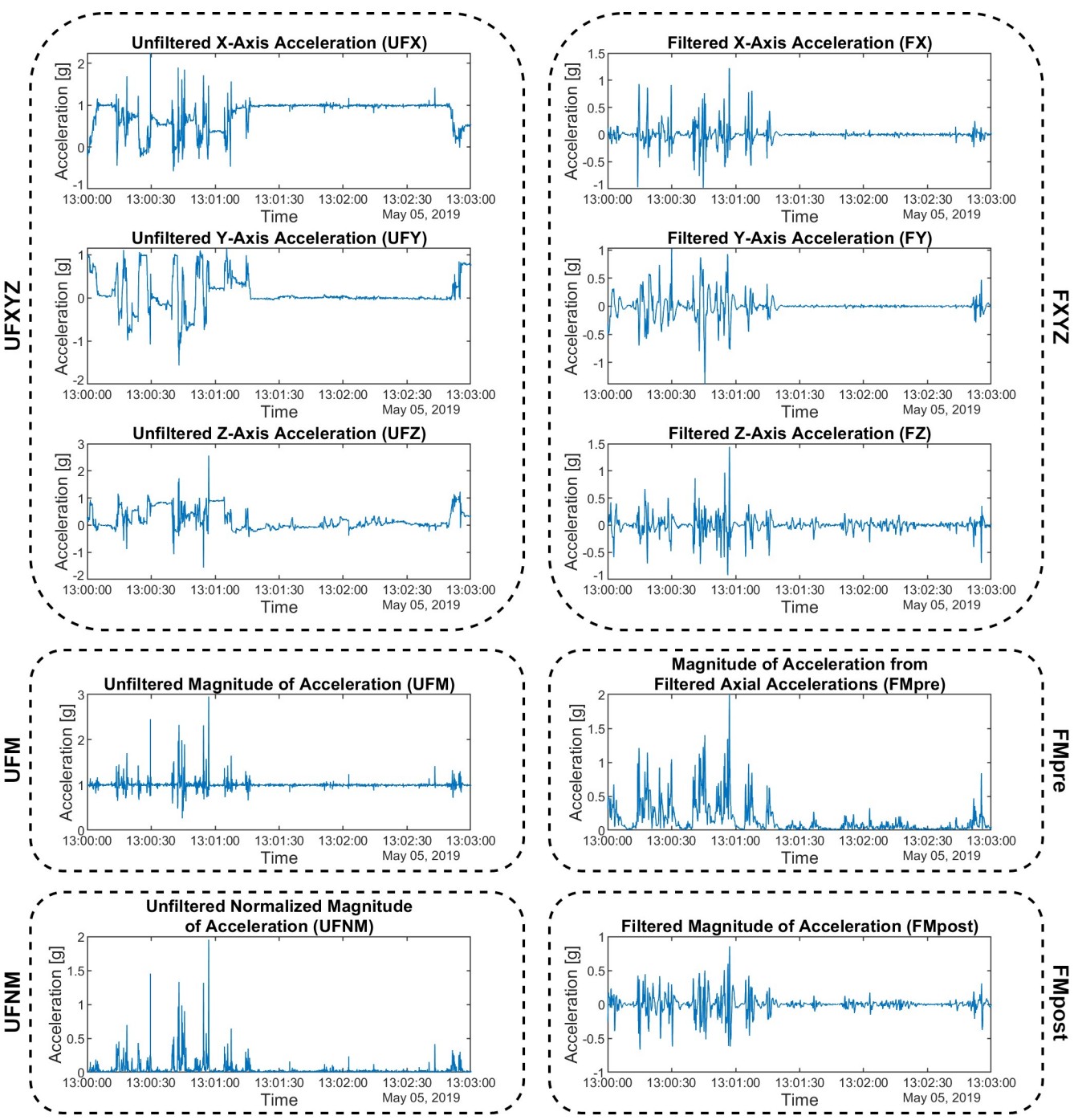

**Fig 3. Examples for the 6 different dataset types.** UFXYZ and FXYZ datasets are compositions of the axial sub datasets. The waveforms cover the same 3-minute-long section of a subject's motion.

examined below. Fig 2 shows a raw acceleration signal measured on the x-axis and the waveform filtered by the 3rd order filter.

FXYZ denotes the data set obtained by conditioning the UFX, UFY, UFZ signals with a digital Butterworth bandpass filter [15] ($f_L$ = 0.25 Hz, $f_H$ = 2.5 Hz, order 3), referred to as FX, FY, FZ per axis (Fig 3).

If necessary, *g* can be eliminated without filtration [21] in the case when we are working with the magnitude of acceleration as shown in Eq 1. Let UFM be the magnitude of the acceleration signal defined by the unmodified vector components (UFXYZ), and UFNM is a data set composed of normalized values of the UFM data set. In contrast to the filtering-based preprocessing, with normalization, we can eliminate the *g* without the risk of losing those low frequency components of the hand movement that are in the same frequency range as the gravity of Earth (i.e., below 0.25 Hz).

$$UFNM[k] = |\sqrt{UFX[k]^2 + UFY[k]^2 + UFZ[k]^2} - 1\,\mathrm{g}| \tag{1}$$

If we want to get a filtered magnitude of the acceleration signal from triaxial data, we can perform it in two ways [22–24]. The signals measured on the three axes are filtered first, and then the resultant magnitude is calculated, or the resultant acceleration signal obtained from the raw axial signals is filtered. Since the former approach is common for most applications, but the latter case is simpler and less operation-intensive (as only one digital filtering needs to be applied), both methods were used in our comparative study. FMpost is the filtered magnitude of acceleration defined by the UFM dataset filtered by the presented filter, while FMpre is the filtered magnitude of acceleration defined by the vector components (FXYZ) filtered by the x, y, and z axial bandpass filters.

A total of 6 types of input datasets can be generated, to which different activity metrics can be applied:

- Unfiltered axial accelerations (UFXYZ)

- Unfiltered magnitude of acceleration (UFM)

- Unfiltered normalized magnitude of acceleration (UFNM)

- Filtered axial accelerations (FXYZ)

- Magnitude of acceleration from filtered axial accelerations (FMpre)

- Filtered magnitude of acceleration (FMpost).

The 6 types of input datasets and a comparison of the magnitudes of acceleration calculated in 4 different ways are shown in an example in Fig 3. As it can be seen, the constant acceleration *g* disappeared in the case of FMpost, while in the case of UFNM and FMpre, the waveform was also modified by taking the absolute value in the calculation.

## Activity metrics

An activity signal can be calculated from one of the datasets (obtained as the output of the preprocessing) using an activity metric. We compared 7 significantly differently calculated metrics from the actigraphy literature; they are summarized in Table 1.

In the scientific field, the most widely used actigraph devices in the literature [18,36,37] are manufactured by Ambulatory Monitoring, Inc. among others [34] and are capable of using three different activity definitions [38]: Proportional Integration Method (PIM), Zero Crossing Method (ZCM) and Time Above Threshold (TAT). Due to the widespread use of these actigraphs, these metrics can be considered as standard activity calculation methods in the field of medical applications. The PIM mode deserves special attention as it is mainly recommended for the characterization of activity in research on sleep medicine [39].

No information can be found on the details of the implementation of the integration for PIM, so in addition to the simplest numerical summation, we also used Simpson's 3/8 Rule

Table 1. The 7 activity metric definitions from the actigraphy literature compared in this study.

| Activity metric | Definition |
|---|---|
| Proportional Integration Method (PIM) | PIM integrates the acceleration signal for a given epoch. In the following, we use the simplest numerical integration (Riemann sum): $$PIM = T_s \sum_{i=1}^{n} x_i,$$ where $x_1, x_2, \ldots, x_{n-1}, x_n$ are the $n$ acceleration values of the given epoch, and $T_s$ is the sampling time of the acceleration signal. |
| Zero Crossing Method (ZCM) | ZCM counts the number of times the acceleration signal crosses a $T_{ZCM}$ threshold for each epoch. |
| Time Above Threshold (TAT) | TAT measures the length of time that the acceleration signal is above a $T_{TAT}$ threshold for each epoch. |
| Mean Amplitude Deviation (MAD) | $$MAD = \frac{1}{n} \sum_{i=1}^{n} |r_i - \bar{r}|,$$ where $r_1, r_2, \ldots, r_{n-1}, r_n$ are the $n$ magnitude of acceleration values of the given epoch, $\bar{r}$ is their arithmetic mean. |
| Euclidean Norm Minus One (ENMO) | $$ENMO = \frac{1}{n} \sum_{i=1}^{n} \max(r_i - 1, 0),$$ where $r_1, r_2, \ldots, r_{n-1}, r_n$ are the $n$ magnitude of acceleration values of the given epoch. The values of $r$ are in $g$. |
| High-pass Filtered Euclidean Norm (HFEN) | $$HFEN = \frac{1}{n} \sum_{i=1}^{n} r_{fi},$$ where $r_{f1}, r_{f2}, \ldots, r_{fn-1}, r_{fn}$ are the $n$ magnitude of filtered acceleration values of the given epoch. Earth's gravity is already eliminated from the axial accelerations by high-pass filtering before magnitude calculation. |
| Activity Index (AI) | $$AI = \sqrt{\max\left(\frac{1}{3}\left(\sum_{m=1}^{3} \sigma_m^2 - \bar{\sigma}^2\right), 0\right)},$$ where $m = 1, 2, 3$ corresponds to the three axes, $\sigma_m^2$ is the variance of the vector components along the $m$th axis of the epoch, and $\bar{\sigma}^2$ is the variance of the baseline noise of the total measurement data (*systematic noise variance*). |

A brief description of their working principles is indicated next to their names.

method, but the obtained activity signals showed almost perfect agreement (see S1 Appendix), therefore in the following, we will only discuss the case of the simplest implementation.

While PIM characterizes the intensity of motion, ZCM is related to the frequency of motion. ZCM compares the acceleration values with a threshold level in a given epoch. The activity value determined by the method for a time slice is equal to the number of threshold level intersections. Contrary to the name of the method, the threshold level should be chosen not for 0, but for a slightly higher value above the noise level; it has no universal value.

TAT is the time spent moving in a given epoch, i.e., the active time. Like ZCM, it compares magnitude values with a threshold level, but instead of measuring the number of level intersections, it measures the time duration when the acceleration was greater than the threshold level. Like ZCM, the value of the threshold level used by the manufacturer is unknown, and there is no generally accepted value in the literature, only a single recommendation for 0.15 g in a patent without further explanation [23].

Various devices manufactured by ActiGraph, LLC. [32,40–42] and equipped with one-, two- and three-axis accelerometers are very common in the field of physical activity testing. The tools provide a so-called Activity Count (AC), which is a proprietary activity metric, so with the advent of other vendors, AC has become a collective term that can be supported by different algorithms [7,43]. However, according to the manufacturer's description [44,45], the

AC metrics of ActiGraph devices are very similar to PIM, which performs numerical integration, so the presented analyses are limited to PIM.

The following metrics have been proposed in other publications [46–48]: Mean Amplitude Deviation (MAD), Euclidean Norm Minus One (ENMO), High-pass filtered Euclidean Norm (HFEN) and Activity Index (AI).

The ENMO [47,49] method divides the sum of the acceleration values in *g* greater than 1 (i.e., greater than *g*) by the number of values in the epoch. This method can only be applied to data series from which the effect of gravity of Earth has not yet been eliminated, as this is done by the method itself.

The MAD [47] determines the average distance of the magnitude values in a given epoch from its own mean value. Similarly to ENMO, this method eliminates the effect of the gravity of Earth by definition. However, while ENMO can only be used on magnitude data, where Earth's gravity is not eliminated, there is no such restriction in case of MAD. Also, though by definition MAD applies to the magnitude of acceleration vector, there is no technical barrier to applying it per axis, so we examined this option in the following, too.

The HFEN [46] determines the average of the magnitude of acceleration values. These magnitude values are calculated by preprocessed triaxial accelerations. During the preprocessing, a 4th order Butterworth high-pass filter with 0.2 Hz cut-off frequency is applied. Since the gravity of Earth is already eliminated at the filtering process, it is not needed to subtract 1 g from the $r_i$ values, as we have seen at the ENMO method. In the following comparisons, the filter with the settings described here is also applied to HFEN when the 3rd order filter described earlier is applied to the other metrics.

The AI determines the activity value using the variance of the vector components and a so-called systematic noise variance. The latter can be determined from the sections of the measurement data when the measuring device has not moved, since this is nothing, but the sum of the variances of such sections is taken per axis. The introduction of AI was prompted by the incoherent activity literature [48]; however, the authors aimed to introduce metrics suitable for distinguishing motion types, so the purpose of the indicator—and, as we shall see, its operation—significantly differs from that of classical metrics.

As we can see, in each case, a given metric is applied to a given dataset, so in the following, we will denote the metric as an operation and the dataset as its argument. For example, the activity calculated by the PIM method for unfiltered normalized magnitudes of acceleration is PIM(UFNM).

## Processing triaxial data

There are several (typically older) devices that record acceleration values on only one or possibly two axes. There is also considerable literature on the analysis of activities calculated for data per axis with several showing that physical activity can also be well-estimated by analysing the vertical axis only [50].

If we record the data along three axes, they are also handled in a very diverse way for different tools and methods, and it is very difficult to adjust the existing nomenclature. Metrics most common in sleep medicine, like PIM, ZCM and TAT are determined in two ways for triaxial data. In uniaxial mode, the metrics can be applied separately to the axial vector components; for example, as it is described in the user manual of the ACTTRUST actigraph, manufactured by Condor Instruments Ltd. [25]. However, the way of obtaining only one activity value characteristic of a given epoch from the values per 3 axes is not defined by the manufacturer.

On the other hand, in triaxial mode, the magnitude of the acceleration vector can be determined from the acceleration signals measured on the three axes, and the metrics are calculated

based on this magnitude signal [2,22,23]. An example of this is shown by Eq 2, where the triaxial value of PIM is calculated by the simplest numerical integration of magnitude.

$$PIM(UFM) = T_s \sum_{i=1}^{n} \sqrt{UFX[i]^2 + UFY[i]^2 + UFZ[i]^2} \qquad (2)$$

where $n$ is the number of the measured values in an epoch, and $T_s$ is the sampling time.

In contrast, for the AC of the ActiGraph tool, which is popular in other applications, the metrics are applied for the axial signals separately, and in the case of triaxial data, one activity value ("VM3") is determined by using the three ACs according to Eq 3 [48,51,52] in the case of UFXYZ data set.

$$VM3 = \sqrt{AC(UFX)^2 + AC(UFY)^2 + AC(UFZ)^2} \qquad (3)$$

As mentioned, the method of calculating AC is also based on a numerical integration within an epoch, so if we replace AC calculation with a summation, it is easy to see that it does not match Eq 2.

As can be seen, in addition to the fact that different devices use different metrics and preprocessing, the activity calculation procedure also differs in how a final metric is determined from the axial measurements and for which signals and at which processing step the metric is calculated. To examine the effect of this, we used both of the former approaches. On the one hand, the metrics are applied to the triaxial data by applying them to the magnitude of the acceleration signal calculated from the filtered or unfiltered axial signals. On the other hand, we apply metrics to uniaxial data too, so we apply them to the triaxial signals separately. These metrics per axis are also compared with the triaxial results on their own; furthermore, a final activity indicator is calculated from the three indicators obtained for the three axes in a number of different ways, such as by their sum, sum of squares or VM3 in Eq 3.

## Methods of examination

It can be seen that the obtained activity signal depends on the preprocessing of the acceleration data, as well as on the activity metrics used and on how it is applied to the axial measurements. Our aim is to examine in detail how and to what extent they depend on these steps. Based on the different preprocessings and activity metrics used, we determined all possible (a total of 148) minutely-sampled activity signals from a single subject's 10-days-long raw acceleration data. Then we computed the Pearson's correlation coefficients between these minutely-sampled activity signals to assess their similarity [8,50], which resulted in a 148×148 correlation matrix. It was performed on all the 42 subjects to obtain 42 correlation matrices. Then we reduced these matrices to a single, averaged matrix by calculating the mean and standard deviation of the identically located cells of the 42 matrices. By analysing the averaged correlation matrix, we could observe how the different activity calculation procedures averagely correlate with each other, based on 42 different subjects' 10-day-long recordings. This comparison was performed over time- and frequency-domain. In order to do this, we also had to examine some issues related to the application and implementation of the activity calculation methods, which are presented below.

Not all metrics can be applied to all the 6 datasets presented earlier, as detailed and demonstrated in the S2 Appendix for each metric and dataset. In summary, PIM, ZCM and TAT are inapplicable on the UFXYZ dataset because, for one axis, the measured acceleration caused by the gravity of Earth depends on the orientation of the device. Since we do not have information about the orientation, these effects of the presence of the unknown amount of the $g$ in the

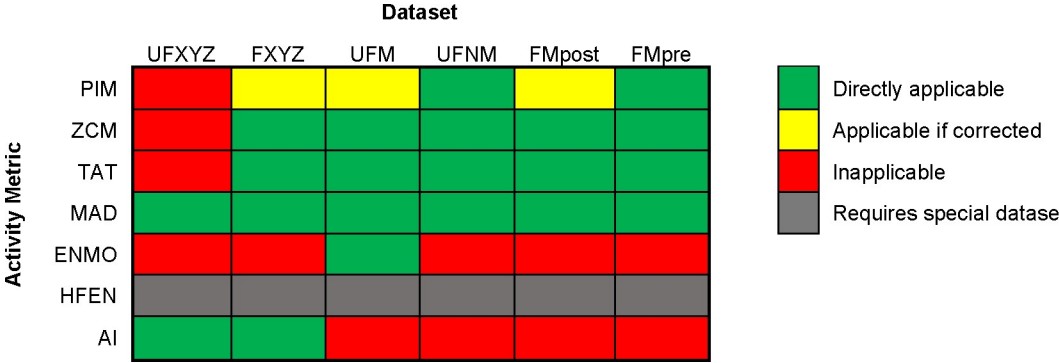

**Fig 4. Summary of metrics' applicability for the different datasets.** Details are presented in S2 Appendix.

measurements cannot be corrected without low-pass filtering. If no acceleration other than the gravity of Earth acts on the actigraph, then the length of the magnitude acceleration vector should be 1 g, therefore the presence of gravity of Earth can be handled for the other datasets. In addition, PIM can only be applied with further technical corrections (e.g., taking the absolute values) to the FXYZ and FMpost datasets due to the integration of negative acceleration values and to the UFM datasets because of the presence of gravity of Earth. Owing to the negative acceleration values in the FXYZ and to the fact that the FMpost dataset moves around 0 g by filtering, the question arises if further correction may be needed in the case of the ZCM and TAT calculations as well, which will be examined in more details later.

According to their definition, AI can only be applied to uniaxial datasets (UFXYZ and FXYZ), while ENMO is applicable only on that dataset which is built up by the magnitudes of the resultant acceleration vectors and which contains the gravity of Earth (UFM). As mentioned in its introduction, HFEN requires a specially conditioned dataset. MAD is the only metric that can be applied to each dataset (if its definition is extended to axial signals). The summary of metrics' applicability for each dataset is shown in Fig 4.

In the case of two classical metrics, it is necessary to use level crossing. Unfortunately, there is no satisfactory description in the literature on how these should be chosen, so we examined the question in detail. The ZCM equates the activity value of an epoch of acceleration signal with the number of level intersections taken at a single $T_{ZCM}$ threshold level, while the TAT gives the total time of the values above the $T_{TAT}$ threshold level. The threshold levels should be placed above the measurement noise level, otherwise high-frequency noise may generate false activity values. In order to determine the ideal value of the threshold levels, starting from the smallest possible $T_{ZCM}$ and $T_{TAT}$ value and taking cumulative steps of 0.05 g, different threshold values were determined for the given dataset type. Correlation coefficients were calculated between the activity signals produced with these different threshold levels and between these signals and activity signals based on other metrics.

Another factor on which the shape of the activity signal depends, and which has not been discussed here yet, is the size of the epochs for which the activity metric is applied, i.e., how long time slots are processed for a characteristic activity value due to the data reduction. Fortunately, this parameter has been investigated by several studies in the past [53]. Epoch length can also be application- and metric-dependent, typically ranging from a few seconds to a minute [54]. Since the 1-minute epoch length is very common, and it has been shown that substantial extra information can no longer be obtained at a higher resolution [55,56], we used the value of $T_e$ for 60 seconds below. In the following, uniformly with this epoch length, the analyses shown will be performed on 42 10-day-long measurements.

### Ethics statement

The study was carried out as a part of research entitled "Examination neurobiological, cognitive and neurophenomenological aspects of the susceptibilities to mood swings or unusual experiences of healthy volunteer students„, and was approved by the Human Investigation Review Board, University of Szeged, Albert Szent-Györgyi Clinical Centre, Hungary (No 267/2018-SZTE) following its recommendations. All subjects gave written informed consent under the Declaration of Helsinki was informed of their right to withdraw at any time without explanation and they were financially compensated.

## Results and discussion

Using the examination methods presented earlier, we firstly determined the appropriate thresholds required to calculate ZCM and TAT metrics. Once we have determined how to implement each metric calculation, we mapped the effect of different preprocessings, different treatment methods of axial measurements and the similarity of different metrics.

Correlation analysis was used for both investigations. In each case, when examining the correlation of two signals calculated in some way, we proceeded as follows: for the motion signal of a given subject, the two activity signals were determined according to the methods to be compared, and then their correlation coefficient was calculated. This was done for all 42 subjects, and the means and standard deviations of these correlation coefficients were used in the analysis.

### Determination of threshold levels

To determine the optimal value of the $T_{ZCM}$ and $T_{TAT}$ thresholds, we calculated the ZCM and TAT activity signals using different threshold levels and then examined their relationship to activity signals based on other metrics. According to Fig 4, the ZCM and TAT methods can be applied to 5–5 different datasets. Therefore, the analysis was performed for each dataset.

It can be seen in Fig 3 that the acceleration values of the FXYZ and FMpost datasets are centralized around 0 g. The question arises in the case of these datasets whether we have to apply an additional negative threshold level due to the negative acceleration values, or equivalently, whether we have to examine the level crossing of the absolute values of the two datasets. As it is detailed in S3 Appendix, we presented that the activity signals obtained from the original and the full-rectified data series show an almost perfect correlation for both metrics. Besides that, the full-rectification approximately doubles the generated activity values as it is expected for datasets that exhibit certain symmetricity around 0 g. Therefore, ZCM and TAT metrics are directly applicable to all of the mentioned 5 datasets using only one positive threshold level.

To find the appropriate value of this threshold, we incremented it using cumulative steps. Since UFM is the only dataset where the gravity of Earth is not eliminated, $T_{ZCM}$ and $T_{TAT}$ were increased from 1 g. For every other dataset, we increased its value from 0 g.

Activity signals obtained with different threshold levels were compared with the ENMO and HFEN activities, as they can only be calculated in one way (see Fig 4), thus avoiding the examination of additional dependent variables during the comparison. Figs 5 and 6 indicate the results for a dataset containing vector magnitudes (UFM) and a dataset containing axial accelerations (FY) as examples, but SFig 3 of the S3 Appendix provides additional graphs for the remaining datasets.

For both the ZCM and TAT activity signals calculated for each dataset, it can be observed that the correlation with other metrics increases rapidly with increasing $T_{ZCM}$ and $T_{TAT}$ and reaches a value above 0.85, which suggests a strong relationship, and with further increase of

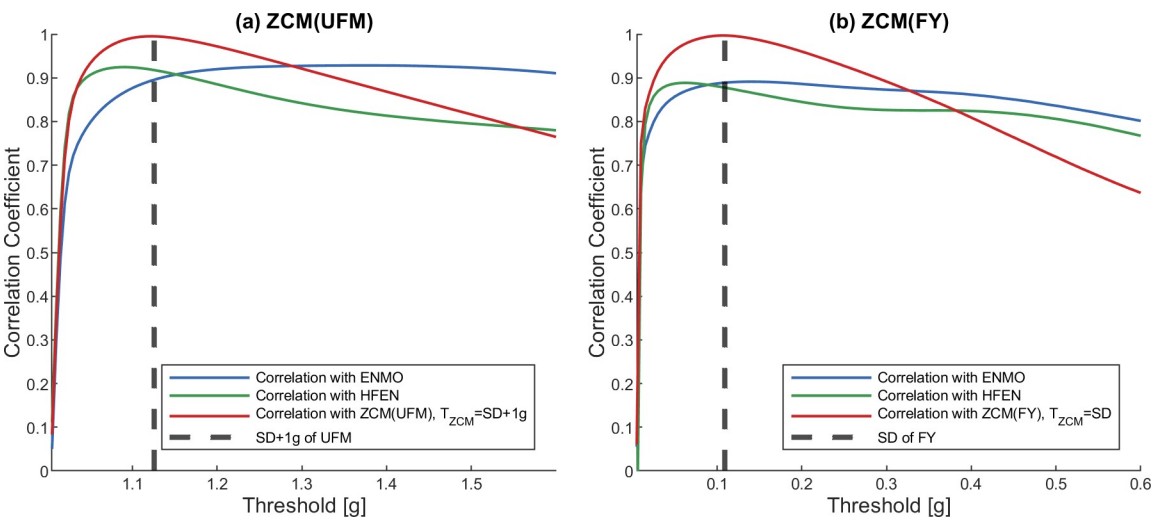

**Fig 5.** Correlation between ZCM activity signal and activity signals based on other metrics using different TZCM thresholds in the case of UFM (a) and FY (b) datasets. The datapoints of the correlation curves as well as the SD of the datasets were based on the mean of 42 measurements. On each panel, the correlation curves show the Pearson's correlation coefficients between the ZCM activity signals calculated with the given TZCM threshold value (x-axis) and the ENMO activity signal, the HFEN activity signal, the ZCM activity signal calculated using the SD+1 g (a) or SD (b) of the dataset as threshold.

threshold levels, it decreases with different rates. If $T_{\mathrm{ZCM}}$ and $T_{\mathrm{TAT}}$ are set to the standard deviation (SD) of the dataset, a strong correlation can be observed with both ENMO and HFEN activity signals. Note that in the case of UFM, it is necessary to add 1 g to the SD due to the constant 1 g present owing to the gravity of Earth.

On each figure, the red correlation curves represent the correlation between the ZCM/TAT activities calculated with the given $T_{\mathrm{ZCM}}/T_{\mathrm{TAT}}$ threshold value and the same activity metric calculated using SD of the dataset (or SD+1 g in the case of UFM) as the threshold. Based on the

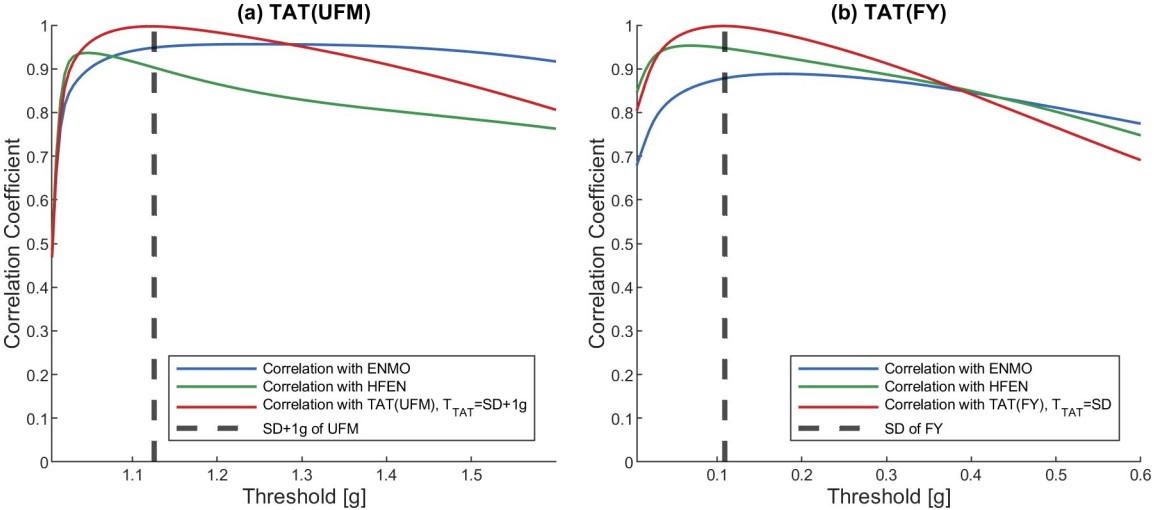

**Fig 6.** Correlation between TAT activity signal and activity signals based on other metrics using different TTAT thresholds in the case of UFM (a) and FY (b) datasets. The datapoints of the correlation curves as well as the SD of the datasets were based on the mean of 42 measurements. On each panel, the correlation curves show the Pearson's correlation coefficients between the TAT activity signals calculated with the given TTAT threshold value (x-axis) and the ENMO activity signal, the HFEN activity signal, the TAT activity signal calculated using the SD+1 g (a) or SD (b) of the dataset as threshold.

figures, it can be concluded that the correlation just marginally changes in the region around the SD of the datasets, i.e., in this range of thresholds, the calculated activity signals are very similar. It is important to note that the only recommendation we have for the TAT method threshold is 0.15 g [23], which in all cases falls within this small range around the mentioned SD.

In addition, since using the SD of the measured dataset is an adaptive method that can have several advantages over a pre-fixed threshold value, it can be used as a universal threshold level for the ZCM and TAT methods. Therefore, in the further application of the ZCM and TAT metrics, the value of $T_{ZCM}$ and $T_{TAT}$ was dynamically set to the SD of the given input data (in the case of UFM, it is necessary to add 1 to the SD due to the constant 1 g present owing to the gravity of Earth).

## Comparison of different activity calculations using correlation analysis

Activity metrics were run for all possible preprocessed dataset types that could be generated from a given actigraphic measurement, and correlations were calculated between these activity signals. Activity calculations were also performed for the resulting vector and separately for the axes. We also tested additional indicators which were generated as different compositions of activity values obtained for the axial data. The correlation analysis was also performed in the time and frequency domain. 148 different types of activity signals were generated for each subjects' measured motion; the correlation of each activity signal with every other and itself was calculated. The S1 Table contains the means and standard deviations of the correlation coefficients calculated for the 42 subjects' motion.

Based on the values of the correlation coefficients, it became possible to quantify the extent of the temporal and spectral similarities of the different activity signals. In the following, we examine these relationships by analysing parts of the S1 Table.

**Effect of preprocessing.** Out of the 7 examined metrics, the ENMO method is compatible with only one type of preprocessed dataset, which is unfiltered and normalized, and HFEN requires a specially conditioned dataset that is filtered and not normalized, so we examined the effect of filtering and normalization on the other 5 metrics. The correlations between the activity signals calculated using the same metric, but different preprocessing methods are shown in Table 2 below. In the table, only the different preprocesses of the magnitude of the acceleration signal are compared, the calculation of activity from the axial data is examined in more details later.

Table 2A shows a comparison of activity signals calculated from normalized (UFNM) and raw data (UFM) in the case of the 4 possible metrics. It can be seen that with the exception of PIM, it is true for the other metrics that the effect of normalization is negligible since the smallest correlation between the raw and normalized signals is 0.97. In the case of PIM (where further processing on datasets was required, see S2 Appendix), the table indicates that although there is a strong similarity, the difference is larger than in the other metrics. Therefore, in the following, if we examine the unfiltered signals in the analysis, they will be examined only for the UFM dataset in the case of ZCM, TAT and MAD, and for the UFM and UFNM datasets for PIM.

In Table 2B, we can see that the filtering does not cause a significant difference in the case of AI; the correlation between the activities calculated for UFXYZ and FXYZ datasets is strong. In Table 2A, we can also examine the effect of both types of filtering presented previously. In the case of vector magnitudes, the activity signals calculated for FMpost show a very high similarity to the activity signals calculated for the unfiltered signals, regardless of the metric. However, the table also reveals that in the case of FMpre, when we filter the axial accelerations

**Table 2. The effect of the normalization and filtering presented by the correlations of the activity signals calculated using the same metric, but different preprocessing methods.**

| (a) | **PIM** | **UFM** | **UFNM** | **FMpre** | **\|FMpost\|** |
|---|---|---|---|---|---|
| | **UFM** | 1±0 | 0.84771±0.04 | 0.73104±0.06 | 0.79153±0.06 |
| | **UFNM** | 0.84771±0.04 | 1±0 | 0.91687±0.02 | 0.9859±0.01 |
| | **FMpre** | 0.73104±0.06 | 0.91687±0.02 | 1±0 | 0.91655±0.02 |
| | **\|FMpost\|** | 0.79153±0.06 | 0.9859±0.01 | 0.91655±0.02 | 1±0 |
| | **ZCM** | **UFM** | **UFNM** | **FMpre** | **FMpost** |
| | **UFM** | 1±0 | 0.97379±0.01 | 0.85344±0.03 | 0.97519±0.01 |
| | **UFNM** | 0.97379±0.01 | 1±0 | 0.89848±0.02 | 0.95753±0.02 |
| | **FMpre** | 0.85344±0.03 | 0.89848±0.02 | 1±0 | 0.84449±0.04 |
| | **FMpost** | 0.97519±0.01 | 0.95753±0.02 | 0.84449±0.04 | 1±0 |
| | **TAT** | **UFM** | **UFNM** | **FMpre** | **FMpost** |
| | **UFM** | 1±0 | 0.98971±0 | 0.92047±0.02 | 0.98005±0.01 |
| | **UFNM** | 0.98971±0 | 1±0 | 0.9317±0.01 | 0.99175±0 |
| | **FMpre** | 0.92047±0.02 | 0.9317±0.01 | 1±0 | 0.93364±0.01 |
| | **FMpost** | 0.98005±0.01 | 0.99175±0 | 0.93364±0.01 | 1±0 |
| | **MAD** | **UFM** | **UFNM** | **FMpre** | **FMpost** |
| | **UFM** | 1±0 | 0.97592±0.01 | 0.78673±0.04 | 0.98676±0.01 |
| | **UFNM** | 0.97592±0.01 | 1±0 | 0.86405±0.04 | 0.94755±0.02 |
| | **FMpre** | 0.78673±0.04 | 0.86405±0.04 | 1±0 | 0.7685±0.05 |
| | **FMpost** | 0.98676±0.01 | 0.94755±0.02 | 0.7685±0.05 | 1±0 |
| (b) | **AI** | **UFXYZ** | **FXYZ** | | |
| | **UFXYZ** | 1±0 | 0.90292±0.02 | | |
| | **FXYZ** | 0.90292±0.02 | 1±0 | | |

The Pearson's correlation coefficients are calculated for 42 measurements, and their means and SDs are represented. Sub table (a) reveals the correlation between the activity signals calculated by those metrics which are applicable on the raw, filtered and also on the normalized datasets containing the magnitudes of acceleration. On the other hand, sub table (b) shows the correlation between the activity signals calculated by the AI activity metric which is applicable on the raw and filtered datasets.

before the magnitude calculation, the effect of filtering cannot be ignored. For ZCM and TAT, the correlation is still strong (0.84 is the lowest correlation value), but for PIM and MAD, although the similarity is clear, we can observe a significant difference between the filtered and unfiltered signals. Furthermore, it can be stated in general that the difference between activities based on raw (UFM) and filtered (FMpre) data is larger than between the metrics applied on normalized (UFNM) and filtered (FMpre) data; however, this difference is not significant.

On the whole, it can be concluded that for further examinations, both filtered and unfiltered signals need to be examined separately.

**Comparison of activity metrics.** Our main goal was to describe the relationship between different activity metrics. Once we know which metrics are worth comparing for which preprocessed datasets, and we are aware of the optimal settings for the metrics, we have the opportunity to compare the activity signals obtained with different metrics.

Figs 7 and 8 present a comparison of the activity signals calculated using different metrics in time-domain. The input of the AI method is the unfiltered or filtered acceleration measured on the three axes (UFXYZ, FXYZ), but for the other metrics, the activity signals were calculated for the unfiltered and two types of filtered magnitudes of acceleration (UFM and FMpost in Fig 7, UFM and FMpre in Fig 8). In accordance with the above, two types of unfiltered datasets, raw and normalized datasets (UFM and UFNM) were also examined for PIM. As ENMO

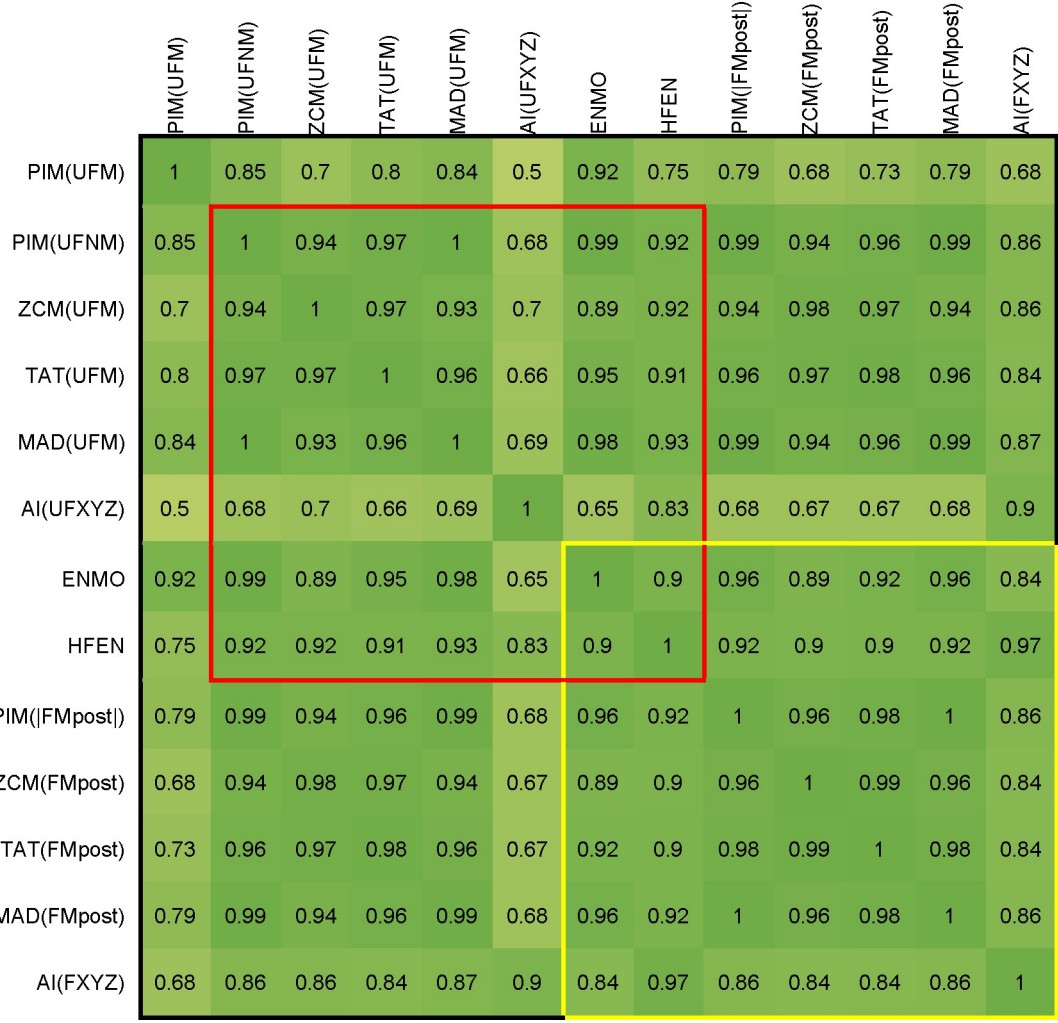

**Fig 7. Correlation between activity signals calculated from raw (UFM, UFXYZ) and filtered datasets (FMpost, FXYZ).** In addition, we included the PIM metric applied on the UFNM dataset, too. The Pearson's correlation coefficients are calculated by 42 measurements, and their means are represented. Cells in the red square indicate correlations between activity signals calculated by the unfiltered datasets, and cells in the yellow square include correlations between activity signals calculated by the filtered datasets. Since our goal is to compare all 7 metrics, but no filtering is possible in the case of ENMO, and as HFEN demands a specially filtered dataset, we included them in both comparisons.

and HFEN can only be calculated in one way, in their cases, it is not possible to compare the obtained activities for the filtered and unfiltered data.

It can be clearly seen in Figs 7 and 8 that the activity signals calculated using different metrics generally show strong similarities for both unfiltered and filtered datasets. In the case of PIM, it can be observed that the signals calculated for the normalized dataset (UFNM) show a much stronger correlation with the other metrics than the raw dataset (UFM), so PIM (UFNM) was included in the comparisons.

Fig 7 shows a comparison of the activity signals calculated for the unfiltered dataset (UFM) using different metrics with a red square. Comparison of the activity signals calculated for the filtered magnitude data (FMpost) is marked with a yellow square. Fig 8 presents a similar comparison between activity signals determined for UFM with a red square, and for vector magnitudes of filtered axial data (FMpre) with a blue square. As mentioned above, in the case of AI

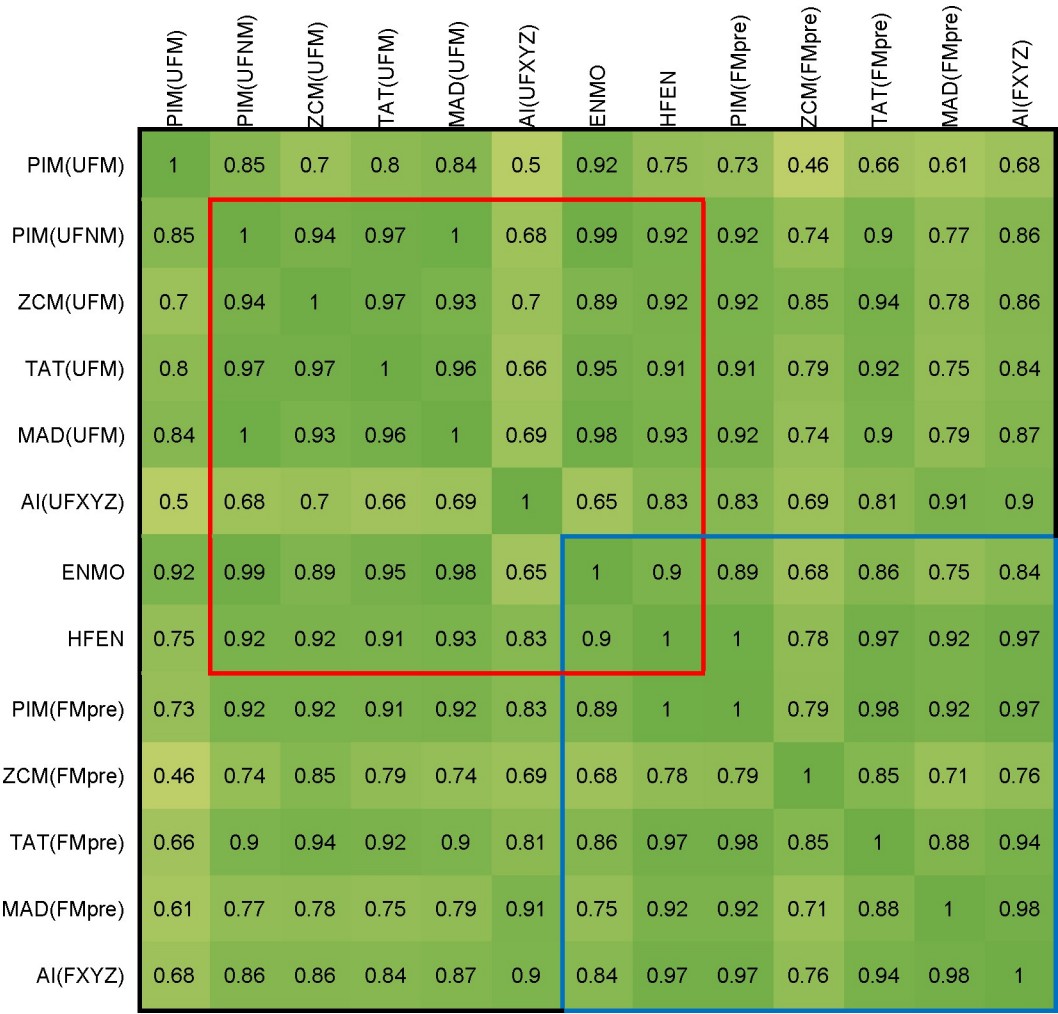

**Fig 8. Correlation between activity signals calculated from raw (UFM, UFXYZ) and filtered datasets (FMpre, FXYZ).** In addition, we included the PIM metric applied on the UFNM dataset, too. The Pearson's correlation coefficients are calculated by 42 measurements, and their means are represented. Cells in the red square indicate correlations between activity signals calculated by the unfiltered datasets, and cells in the blue square include correlations between activity signals calculated by the filtered datasets. Since our goal is to compare all 7 metrics, but no filtering is possible in the case of ENMO, and as HFEN demands a specially filtered dataset, we included them in both comparisons.

metric, UFXYZ and FXYZ datasets were used. Furthermore, since our goal is to compare all 7 metrics, we also added those metrics to the comparison that are not currently applicable to the examined datasets. For this reason, both the HFEN and the ENMO metrics were involved in the comparison. However, since no filtering is possible in the case of ENMO, and as HFEN demands a specially filtered dataset, we included them in both squares.

All activity signals in the red square, except AI(UFXYZ), show a very strong relationship with the other–the correlation value is 0.89 in the worst case. The separation of the AI method is understandable based on the purpose of its introduction: the authors did not look for an indicator that returns the value of classical metrics, but for one that also carries additional information which can be useful in distinguishing between different types of movements [48].

Similarly strong relationships can be seen for most of the activity signals calculated using the filtered datasets, marked with a yellow square (FMpost) in Fig 7 and with a blue square

(FMpre) in Fig 8. For FMpost data in Fig 7, only the AI shows a slightly smaller correlation with the other signals, in which case the difference has been discussed earlier for unfiltered data. For the metrics applied on FMpost dataset, the lowest correlation value is 0.96. Moreover, the HFEN metric with a special filtering and the ENMO metric without filtration show a very strong relationship with the other signals, too.

In the case of FMpre data in Fig 8, the ENMO shows a slightly larger difference, which can be easily explained by the lack of filtering. More surprisingly, the ZCM shows even more significant differences from other metrics, which can be attributed to the sensitivity of the level intersections. The correlation of the other 5 metrics is very strong here, too.

A completely identical pattern is observed in the frequency-domain; however, the differences are smaller. The spectral analysis was based on the values of the spectral correlation coefficients calculated between the power spectrum densities of the activity signals. The corresponding figure can be found in SFig 1 of the S4 Appendix.

The upper right and lower left parts of the tables in Figs 7 and 8 show that the correlation between the activity signals calculated for the unfiltered (UFM) and filtered magnitude (FMpost) data is strong, but much weaker in the case of UFM and vector magnitudes of filtered axial data (FMpre), as we could expect in the light of the previous chapter. This is an important finding, as most actigraphs filter acceleration signals per axis [51,57], and several studies work with FMpre datasets [22,23]. Moreover, many studies provide little and incomplete information on which tools were used and on how and with what parameters the filtering was carried out.

On the whole, based on Table 2 and Figs 7 and 8, we can conclude that TAT alone shows a strong agreement both with all the different preprocessings and with other activities calculated for the data preprocessed in almost any way. Therefore, it can be said that most metrics produce a similar activity signal for datasets preprocessed in the same way. However, these metrics are generally applied to signals preprocessed in different ways, which, as we have seen, can have significant differences between activity signals. As an example, MAD, AI, ENMO are often applied for unfiltered data in the literature [47,48], while PIM, ZCM and TAT methods are applied on data measured by devices using filtering.

**Number of measured axes.** So far, we have used the magnitude of the acceleration for comparison in the case of each metric, except for AI. However, for 4 of the 7 metrics, it is possible to apply the metric to the axial data separately. In fact, we have no other option for devices that measure acceleration in one or two axes only. Thus, two very interesting question emerge: is it sufficient to determine the activity for fewer axes, and if yes, how does that relate to the activity signal determined on the basis of triaxial data?

Although all three directions may play an important role in evaluating the intensity of a movement, if an axis needs to be chosen, or we have a uniaxial device, the measurement is usually performed along the axis which is vertical when the hand is hung.

Based on Table 2 and S2 Appendix, for 4 metrics, we have the option to apply them to the three axes separately, and only the MAD can be applied to the unfiltered, raw axial acceleration signals (UFXYZ). Accordingly, in Fig 9, the examined metrics are applied to the acceleration signals filtered per axis and to the vector magnitude calculated from them. The correlation between the axial and vector magnitude-based activities are shown in the case of the 42 subjects' measured movements.

It can be clearly seen that for PIM and TAT metrics, there is a very strong correlation between the activities obtained for the axial data and the vector magnitude data. In the case of ZCM and MAD, a close relationship can be observed, too. Based on the S1 Table, the activity signals calculated for different axes show a very close relationship with each other for each metric.

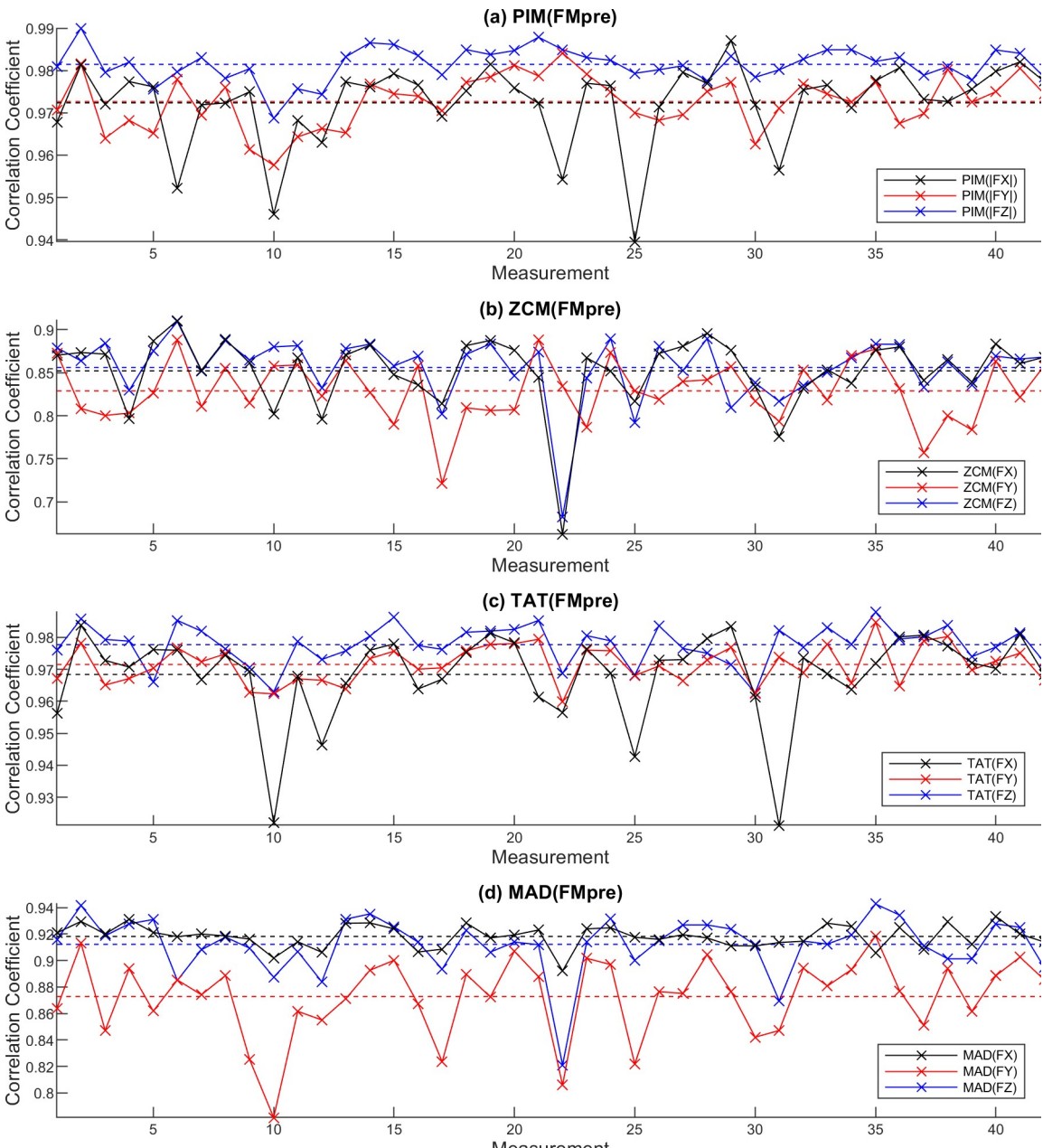

**Fig 9.** Correlation between filtered axial accelerations (FX, FY, FZ) to the vector magnitude calculated from them (FMpre) in the case of 42 measurements regarding the PIM (a), ZCM (b), TAT (c) and MAD (d) activity metrics. The means of the individual correlation curves are also represented as dotted horizontal lines with matching colors.

In the comparison presented in Fig 10, for ease of transparency, the activity signals are only calculated on the FY vertical axis (and UFY in the case of MAD metric) and on the FMpre datasets in the case of all possible metrics.

The cells in the pink square region of the matrix indicate that there is a very strong correlation between the activities calculated for the axial acceleration by all four metrics: PIM, ZCM, TAT and MAD; and the upper right corner of the table (separated by the pink and blue square) show that they are closely related to the activity determined from triaxial data, too.

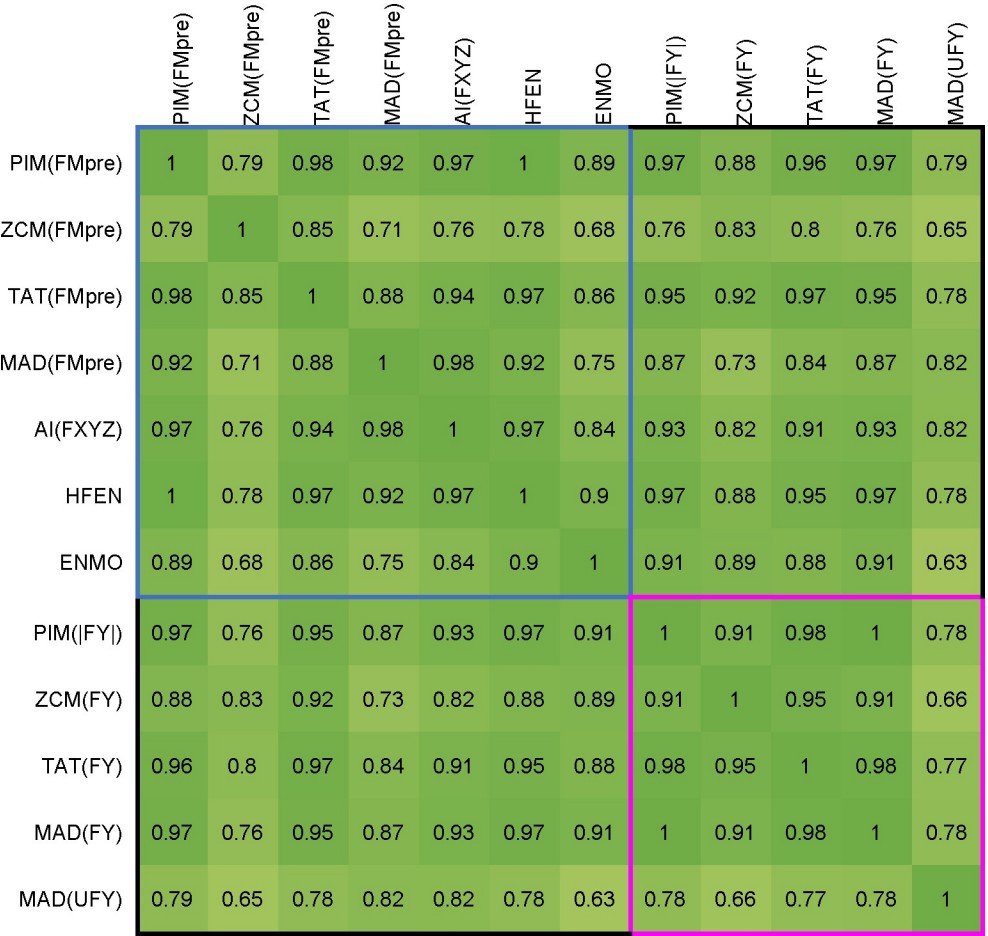

**Fig 10. Correlation between activity signals calculated from filtered Y-axis accelerations (FY) and from vector magnitude of filtered axial accelerations (FMpre).** MAD metric applied to UFY dataset is also included. The Pearson's correlation coefficients are calculated by 42 measurements, and their means are represented. Cells in the blue square indicate correlations between activity signals calculated from the FMpre dataset and AI(FXYZ), ENMO and HFEN signals. Cells in the pink square include correlations between activities calculated from the y-axis datasets for those 4 metrics where we have the option to apply them to the three axes separately.

Interestingly, ZCM reveals a stronger correlation with other metrics per axis than when applied to the magnitude of acceleration. The activity signal obtained for the MAD metric applied to unfiltered y-axis acceleration (UFY) implies a less close relationship with the activities calculated for the filtered y-axis signals (FY) and the activities calculated for the vector magnitudes (FMpre).

The result is consistent with previous studies that compare physical activity with derived indicators and suggest that in many applications there may be a case where it is sufficient to use only one axis, thus measuring only one axis [50,58]. In addition to reduced computational requirements, this reduction can also be significant because the biggest challenges in measuring raw acceleration signals are continuous, high-frequency measurement and data storage for as long as possible. If the data to be measured and stored is reduced by a third, it means a very significant reduction in consumption. At the same time, it is important to validate the result by analysing the signals measured in this way and by comparing the results and indicators obtained from the analysis of the magnitude of the acceleration.

**Further indicators calculated from the activity metrics applied on axial accelerations.**
If we apply the activity metrics on axial data, it is a very relevant requirement to determine one
activity indicator based on the three activity values obtained for the axes. Using this indicator,
we can easily characterize the activity of the subject, as if we had determined one based on the
magnitude of acceleration. There are several ways to do this, for example we can add the values
obtained for the three axes together, we can also sum these values after squaring, or we can
apply Eq 3.

In the following, we determined different new activity indicators using all the 4 possible
axial metrics (PIM, ZCM, TAT and MAD), which we compared with the metrics obtained for
the vector magnitude of acceleration in each case. In the case of the PIM and the ZCM metric,
the comparison is shown in Fig 11.

For the remaining two metrics, the results are presented in SFig 1 of the S5 Appendix. It
can be clearly seen that most of the newly calculated activities (new activity indicators) show a
strong relationship with the activity signals calculated on the basis of the examined 7 metrics,
and a similar pattern can be discovered for all 4 metrics.

It can be stated that overall, the activity metrics determined per axis show a stronger corre-
lation with the vector magnitude-based activity metrics than any tested combinations of the
axial activity signals. An interesting fact is that for all 4 metrics, it is the VM3 as defined in Eq
3 that performs poorly, which indicator is widely used in practice [52,59–61].

Since the activity signals determined per axis also showed a strong correlation with the
activities determined from the triaxial data, it does not seem necessary to look for a new indi-
cator with an additional complex method, their sum or one of the axial activity signals may be
appropriate.

## Conclusion

As presented, the activity signal can be calculated in many ways from the raw actigraphy
recordings. The difference can be caused by the preprocessing of raw acceleration data; i.e.,
whether we normalize, filter, if so, when, and by what metrics we calculate activity and apply it
to the axial signals or to the magnitudes of the resultant vectors. To investigate this, we calcu-
lated all possible 148 activity signals for the triaxial accelerations recorded during the 10-day
movement of 42 subjects and examined the mentioned issues by their correlation analysis.

Since an acceleration signal can be preprocessed in diverse ways before calculating an activ-
ity signal, to standardize the inconsistency found in the literature, we constructed a unified
nomenclature for the datasets, which are created by different preprocessings. We identified
which metrics could be used for each preprocessing method and presented (detailed in the S2
Appendix) what additional correction is required when applying the PIM metric for some
datasets., We examined what threshold value is appropriate for the level crossing based ZCM
and TAT methods, as there is no clear guidance for this in the literature. We found that the
standard deviation of the dataset shows a near-maximal correlation with the other metrics for
each preprocessing and takes a value close to the only recommendation known so far, but it is
adaptive, therefore it appears to be an optimal threshold level.

The effect of possible preprocessing methods was also identified. Removing the gravity of
Earth by normalization only affects the PIM metrics. Our results showed that if we use a filter
instead of normalization, it makes a difference depending on which step of the preprocessing
we apply it on. If the resulting acceleration vectors are filtered, the correlation is strong with
the activity signals calculated from the unfiltered data, but if we filter axially (this is more com-
mon) and then calculate the resultant vector magnitude, the activity calculated from it shows
more significant differences from the previous ones. Based on this, there may be a significant

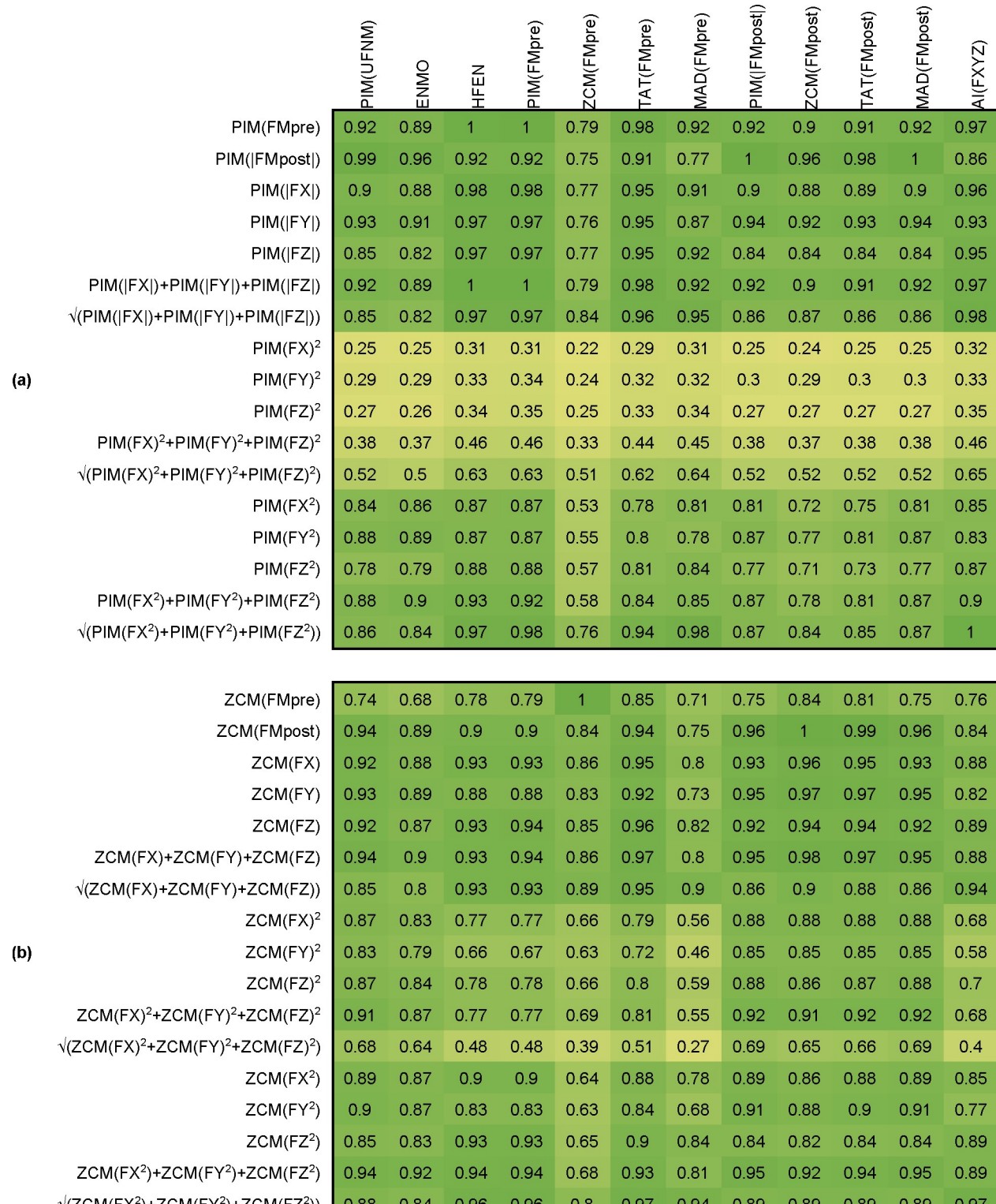

**Fig 11.** Correlation between new activity indicators based on the activity signals determined per axis using PIM (a), ZCM (b) metrics and the 7 metrics applied on filtered vector magnitudes (FMpost, FMpre, FXYZ). Each cell contains the Pearson's correlation coefficient between the two corresponding activity signals. The coefficients are calculated by 42 measurements, and their means are represented.

difference between the metrics widely used for unfiltered signals (e.g., ENMO and MAD) and the activity signals calculated for the filtered signals (e.g., PIM, ZCM and TAT).

In summary, we can state that the relationship between 6 of the examined metrics is strong for the identically prepared data, only the AI shows a more significant difference due to its slightly different purpose. However, the raw signals are usually preprocessed in different ways in the case of different methods and instruments, and based on our results, this may mean more significant differences between the obtained activity signals. Thus, possibly weakly correlated signals are similarly, misleadingly called "activity" in practice, and the results obtained are compared although they are based on differently calculated activity signals.

We have presented that if we apply the metrics for axial acceleration signals–it is possible in the case of 4 of the 7 metrics–, the activity obtained per axis and the combination of activities obtained for the three axes also show a strong correlation with the activity calculated using the magnitude of the acceleration vector. This, being consistent with previous results, shows that in many cases where resource optimization is important, it is enough to measure the acceleration on only one axis. We have also revealed that using the combination of activity values determined per axis is unnecessary, since the combined activity signals show a weaker relationship with the activity signals calculated from magnitude data than the uncombined axial activity signals.

The calculation methods for the 6 similar metrics are simple, there is no significant difference in the complexity of their implementation. In the case of PIM, further technical corrections are required on the data series. In the case of ZCM and TAT, the choice of the appropriate threshold level is important. MAD is the only metric that can be applied on all possible datasets if we extend its usage on axial data, which has no technical barriers. Both ENMO and HFEN can be applied to only one, but a different type of dataset; however, they showed a strong relationship with activity signals calculated with different metrics for both the raw and the two-way filtered datasets (FMpre, FMpost). TAT is applicable on every dataset except for raw acceleration signals (UFXYZ); however, it shows a very strong similarity when applied to both filtered per-axis signals and to differently filtered vector magnitudes, and it also strongly correlates with almost any preprocessed activity signals calculated by other metrics.

## Supporting information

**S1 Table. Correlation matrices.** Our analyses are based on 148×148 time- and frequency-domain correlation matrices. A correlation matrix covers all the possible use cases of every activity metric listed in the article. With these activity metrics and different preprocessing methods, we were able to calculate 148 different activity signals from multiple datasets of a single measurement. Each cell of a correlation matrix contains the mean and standard deviation of the calculated Pearson's correlation coefficients between two types of activity signals based on 42 different subjects' 10-days-long motion. The small correlation matrices presented both in the article and in the appendixes are derived from these $148 \times 148$ correlation matrices. This published Excel workbook contains multiple sheets labelled according to their content. The mean and standard deviation values for both time- and frequency-domain correlations can be found on their own separate sheet. Moreover, we reproduced the correlation matrix with an alternatively parametrized digital filter, which doubled the number of sheets to 8. In the Excel workbook, we used the same notation for both the datasets and activity metrics as presented in this article with an extension to the PIM metric: PIMs denotes the PIM metric where we used Simpson's 3/8 rule integration method, PIMr indicates the PIM metric where we calculated the integral by simple numerical integration (Riemann sum).
(XLSX)

**S1 Appendix. Comparison of the integration methods.**
(PDF)

**S2 Appendix. Applicability of the activity metrics).**
(PDF)

**S3 Appendix. Further details of level crossing analysis.**
(PDF)

**S4 Appendix. Spectral correlation of activity signals.**
(PDF)

**S5 Appendix. Further indicators based on the activity signals determined per axis using TAT and MAD metrics.**
(PDF)

## Acknowledgments

The authors thank Anita Bagi, Szilvia Szalóki, János Mellár and Dénes Faragó for coordinating data collection.

## Author Contributions

**Conceptualization:** Bálint Maczák, Gergely Vadai.

**Formal analysis:** Bálint Maczák, Gergely Vadai.

**Investigation:** Bálint Maczák, Gergely Vadai, András Dér, István Szendi, Zoltán Gingl.

**Methodology:** Bálint Maczák, Gergely Vadai.

**Resources:** István Szendi, Zoltán Gingl.

**Software:** Bálint Maczák.

**Supervision:** András Dér, Zoltán Gingl.

**Visualization:** Bálint Maczák, Gergely Vadai.

**Writing – original draft:** Bálint Maczák, Gergely Vadai.

**Writing – review & editing:** Bálint Maczák, Gergely Vadai, András Dér, István Szendi, Zoltán Gingl.

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
