## [Decision Letter · Decision Letter 0]

30 Sep 2021

PONE-D-21-28857Detailed analysis and comparison of different activity metricsPLOS ONE

Dear Dr. Vadai,

Thank you for submitting your manuscript to PLOS ONE. After careful consideration, we feel that it has merit but does not fully meet PLOS ONE’s publication criteria as it currently stands. Therefore, we invite you to submit a revised version of the manuscript that addresses the points raised during the review process. The two reviewers assessed the manuscript very differently. Reviewer 1. raised some important issues concerning the quality of the manuscript, which should be addressed seriously.

We look forward to receiving your revised manuscript.

Kind regards,

Gábor Vattay, PhD, DSc

Academic Editor

PLOS ONE

Journal Requirements:

Reviewers' comments:

Reviewer's Responses to Questions

**Comments to the Author**

1. Is the manuscript technically sound, and do the data support the conclusions?

Reviewer #1: Partly

Reviewer #2: Yes

2. Has the statistical analysis been performed appropriately and rigorously? 

Reviewer #1: I Don't Know

Reviewer #2: Yes

3. Have the authors made all data underlying the findings in their manuscript fully available?

Reviewer #1: Yes

Reviewer #2: Yes

4. Is the manuscript presented in an intelligible fashion and written in standard English?

Reviewer #1: No

Reviewer #2: Yes

5. Review Comments to the Author

Reviewer #1: This paper examines different accelerometer data pre-processing strategies and describes the correlations between a range of resulting metrics (e.g., MAD, ENMO, zero crossings). The paper is interesting in places, but the rationale for performing the work, and the explicit aims could be better articulated. Because these are underdeveloped, it is unclear what the paper sets out to achieve. The quality of the English could also be improved, as the paper somewhat jarring in places. Ideally it would have been proofread by someone proficient in written English prior to submission (to make it easier for the reviewers to focus on the science). I have some more specific comments below, which may be useful for improving the work:

Introduction

The rationale in the introduction is not well articulated. I feel several parts of the introduction do not contribute towards the rationale and can be removed. E.g., L56–69 describes the failure to report the key aspects of study methodology, and L46-55 is about sleep disorders. The link between these ideas and different accelerometer pre-processing decisions is not clear.

Given the main aim of this study is to compare different metrics obtained from different pre-processing decisions, the authors fail to acknowledge that the vast majority of researchers do not perform any pre-processing themselves – they simply use the output provided by the manufacturer’s software, which in most cases is well documented (e.g., Actigraph, Actical, ActivPAL, Axivity, GENEActiv). What would really strengthen the rationale is a thorough description of how understanding the correlations between metrics derived from raw could be useful to researchers?

The aim(s) could be presented much more explicitly, as there are several parts to the results (e.g., identifying thresholds, correlations, further divided into pre-processing, different activity metrics, axes).

P3L37: What do you mean by ‘actigraph’. Is this the device developed by the Actigraph company? Or broadly speaking a wearable device that contains an accelerometer?

P3L46: I wouldn’t say actigraphy is a method in polysomnogrphy. The validity of using accelerometer data for true measures of sleep quality is also questionable.

P3L54: What do you mean by ‘important results’. This is very ambiguous.

Methods

The headings in this section are somewhat misleading. For example, the ‘Measurement data’ heading is about participants and recruitment, not about the data.

P6L112-118: It sounds like this was a custom-made device. You should provide information on the validity and reliability of the acceleration output and RTC. Is there any drift in the timestamp, and is the sampling frequency constant or does it fluctuate?

P6: A fundamental point here is that given you have used a device that has not been used in any other physical activity research, how can you expect your results to be generalisable other literature?

P6L123: Why were participants with impaired neurocognitive function excluded? This seems strange given the main aim is to compare accelerometer metrics (where greater heterogeneity in your dataset is probably useful).

P6L127: Why was 10 Hz and 8 g chosen? This is a fundamental decision that needs to be justified in relation to the purpose of the study. Why didn’t you use 30 Hz ±6 g or 100 Hz ±8 g which are much more common in physical activity literature. A different sample frequency and measurement range will change the results, which has implications for generalisability.

L144¬, L170, L227 etc: These sections read more like a literature review than a succinct description of the methods, making it difficult to differentiate between theory and what was performed in this study.

P15L329: More information is needed to describe the main analysis (Pearson’s correlation). It is currently unclear what the unit of observation is (i.e., is it the 42 subjects, or is it the epoch data, meaning you have repeated measures for each subject). The number of observations in this analysis is not shown in the results.

Reviewer #2: This is an important paper which should be intensively studied by all the scientists who begin with actigraphic measurements and analysis. It serves both as a critical survey of the methods used in the field and with proposed solutions to cure the deficiencies of the available methods. I propose the publication of the paper after a minor addition about the claim that the gravitational correction can also be done by filtering. I don't really get that because, it seems, in spacial cases, the gravitational component can be in he same frequency range as the activity signal. Example: the human object is rotating his/her wrist in a periodic fashion.

6. PLOS authors have the option to publish the peer review history of their article (what does this mean?). If published, this will include your full peer review and any attached files.

Reviewer #1: No

Reviewer #2: **Yes: **Laszlo Kish

---

## [Author Response · Author response to Decision Letter 0]

9 Nov 2021

First of all, we thank the Reviewers and the Editor very much for reading our manuscript carefully, and for the valuable comments.

According to the Editor's guidance, we have attached a response letter labeled as "Response to Reviewers". In this document, we have responded to each and every point raised by the reviewers. Considering every reviewer comments, we modified the manuscript. Besides the updated version of the manuscript, we also attached a document labeled as "Revised Manuscript with Track Changes", where all of our modifications are highlighted.

---

## [Decision Letter · Decision Letter 1]

29 Nov 2021

PONE-D-21-28857R1Detailed analysis and comparison of different activity metricsPLOS ONE

Dear Dr. Vadai,

Thank you for submitting your manuscript to PLOS ONE. After careful consideration, we feel that it has merit but does not fully meet PLOS ONE’s publication criteria as it currently stands. Therefore, we invite you to submit a revised version of the manuscript that addresses the points raised during the review process.

One of the reviewers sees a major improvement, but still has some objections. I suggest making a final effort and improve the manuscript along the suggested lines. 

We look forward to receiving your revised manuscript.

Kind regards,

Gábor Vattay, PhD, DSc

Academic Editor

PLOS ONE

Journal Requirements:

Reviewers' comments:

Reviewer's Responses to Questions

**Comments to the Author**

1. If the authors have adequately addressed your comments raised in a previous round of review and you feel that this manuscript is now acceptable for publication, you may indicate that here to bypass the “Comments to the Author” section, enter your conflict of interest statement in the “Confidential to Editor” section, and submit your "Accept" recommendation.

Reviewer #1: (No Response)

Reviewer #2: All comments have been addressed

2. Is the manuscript technically sound, and do the data support the conclusions?

Reviewer #1: Partly

Reviewer #2: Yes

3. Has the statistical analysis been performed appropriately and rigorously? 

Reviewer #1: No

Reviewer #2: Yes

4. Have the authors made all data underlying the findings in their manuscript fully available?

Reviewer #1: Yes

Reviewer #2: Yes

5. Is the manuscript presented in an intelligible fashion and written in standard English?

Reviewer #1: No

Reviewer #2: Yes

6. Review Comments to the Author

Reviewer #1: The authors have made extensive revisions to the original submission, and the paper has definitely improved, so well done. In general, I still think the text is overly verbose throughout and could be structured much more succinctly.

One point I will make is I still do not understand the correlation analysis method. It sounds like you obtained a correlation matrix separately for each subject. Specifically, a 148x148 correlation matrix, for each of the 42 subjects (which were later averaged). I don’t know how many observations were used in each correlation. As you had 10 days, did you summarise each of the 148 activity signals per day, so each individual correlation had 10 paired data points? Or were these correlations performed using the raw 10 Hz data (~8.6m data points across 10 days)? The reason why I asked about this in the last review is because a Pearson’s correlation is not an appropriate method of statistical analysis if your observations are related. This should only be used when your observations (i.e., rows in your dataset) are independent. There are other statistical methods that are appropriate for determining association when the data points are related.

Reviewer #2: The Authors successfully addressed my comments. All required questions have been answered and that all responses meet formatting specifications. Thus I propose publication in its present form.

7. PLOS authors have the option to publish the peer review history of their article (what does this mean?). If published, this will include your full peer review and any attached files.

Reviewer #1: No

Reviewer #2: No

---

## [Author Response · Author response to Decision Letter 1]

4 Dec 2021

According to the Editor's guidance, we have attached a response letter labeled as "Response to Reviewers". In this document, we have responded to each and every point raised by the reviewers. Considering every reviewer comments, we modified the manuscript. Besides the updated version of the manuscript, we also attached a document labeled as "Revised Manuscript with Track Changes", where all of our modifications made to the previously revised version are highlighted.

---

## [Decision Letter · Decision Letter 2]

9 Dec 2021

Detailed analysis and comparison of different activity metrics

PONE-D-21-28857R2

Dear Dr. Vadai,

We’re pleased to inform you that your manuscript has been judged scientifically suitable for publication and will be formally accepted for publication once it meets all outstanding technical requirements.

Kind regards,

Gábor Vattay, PhD, DSc

Academic Editor

PLOS ONE

Additional Editor Comments (optional):

Reviewers' comments:

Reviewer's Responses to Questions

**Comments to the Author**

1. If the authors have adequately addressed your comments raised in a previous round of review and you feel that this manuscript is now acceptable for publication, you may indicate that here to bypass the “Comments to the Author” section, enter your conflict of interest statement in the “Confidential to Editor” section, and submit your "Accept" recommendation.

Reviewer #1: All comments have been addressed

Reviewer #2: All comments have been addressed

2. Is the manuscript technically sound, and do the data support the conclusions?

Reviewer #1: Yes

Reviewer #2: Yes

3. Has the statistical analysis been performed appropriately and rigorously? 

Reviewer #1: N/A

Reviewer #2: Yes

4. Have the authors made all data underlying the findings in their manuscript fully available?

Reviewer #1: Yes

Reviewer #2: Yes

5. Is the manuscript presented in an intelligible fashion and written in standard English?

Reviewer #1: No

Reviewer #2: Yes

6. Review Comments to the Author

Reviewer #1: I'm happy with the authors revisions. One suggestion I have is to add the explanation in the authors response (regarding the 14,000 observations) into the manuscript.

Reviewer #2: The paper is fine for publication. This paper has the potential to be a basic reference material for future studies of this kind.

7. PLOS authors have the option to publish the peer review history of their article (what does this mean?). If published, this will include your full peer review and any attached files.

Reviewer #1: No

Reviewer #2: **Yes: **Laszlo Kish

---

## [Editor Report · Acceptance letter]

13 Dec 2021

PONE-D-21-28857R2 

Detailed analysis and comparison of different activity metrics 

Dear Dr. Vadai:

I'm pleased to inform you that your manuscript has been deemed suitable for publication in PLOS ONE. Congratulations! Your manuscript is now with our production department. 

Kind regards, 

on behalf of

Dr. Gábor Vattay 

Academic Editor

PLOS ONE